# Brown Algae Extracts Increase the Tolerance of Tomato Plants to High Temperatures by Improving Morphological, Physiological, Metabolomic, and Transcriptional Parameters

**DOI:** 10.3390/plants14192996

**Published:** 2025-09-28

**Authors:** Oscar Sariñana-Aldaco, Rosa M. Rodríguez-Jasso, Adalberto Benavides-Mendoza, Armando Robledo-Olivo, Pablo Preciado-Rangel, Antonio Juárez-Maldonado, Susana González-Morales

**Affiliations:** 1Program in Science in Protect Agriculture, Universidad Autónoma Agraria Antonio Narro, Saltillo 25315, Coahuila, Mexico; oscarsarinana390@gmail.com; 2Biorefinery Group, Food Research Department, School of Chemistry, Universidad Autónoma de Coahuila, Saltillo 25280, Coahuila, Mexico; rrodriguezjasso@uadec.edu.mx; 3Horticulture Department, Universidad Autónoma Agraria Antonio Narro, Saltillo 25315, Coahuila, Mexico; abenmen@gmail.com; 4Fermentations and Biomolecules Lab, Food Science and Technology Department, Universidad Autónoma Agraria Antonio Narro, Saltillo 25315, Coahuila, Mexico; armando.robledo@outlook.com; 5Horticulture Department, Universidad Autónoma Agraria Antonio Narro, Unidad Laguna, Torreon 27054, Coahuila, Mexico; ppreciador@yahoo.com.mx; 6Botany Department, Universidad Autónoma Agraria Antonio Narro, Saltillo 25315, Coahuila, Mexico; antonio.juarez@uaaan.edu.mx; 7Secretariat for Science, Humanities, Technology and Innovation (SECIHTI), Universidad Autónoma Agraria Antonio Narro, Saltillo 25315, Coahuila, Mexico

**Keywords:** antioxidants, biostimulation, gene expression, heat stress, seaweed, stress tolerance

## Abstract

Currently, biostimulants in the horticultural sector are a tool that is being used to improve the yield and quality of vegetables under optimal and stressful growth conditions. In the present study, we evaluate the effects of foliar application of a hydroethanolic extract of *Sargassum* spp., a commercial extract based on *Ascophyllum nodosum*, and a control with distilled water on growth and biomass, stomatal conductance, photosynthetic pigments, enzymatic and non-enzymatic antioxidants, protein content, and the expression of defense genes in tomato plants (*Solanum lycopersicum* L.) without stress and with high-temperature stress (45 °C). The results showed that *Sargassum* spp. extract only increased the height of tomato plants under stress-free conditions (2.71%) in the last evaluation. The aboveground and total dry biomass of the plants were increased by *Sargassum* spp. extract under stress-free conditions by 9.56 and 8.58%, respectively. Under stress conditions, aboveground dry biomass was increased by 6.66% by *Sargassum* spp. extract. Stomatal conductance, photosynthetic pigments, protein content, enzymatic and non-enzymatic antioxidants, and defense gene expression of tomato plants were positively modified with the use of *Sargassum* spp. and *A*. *nodosum* extract under high-temperature stress conditions. Under stress-free conditions, the described variables were positively modified except for gene expression, where some genes were expressed and others were repressed. The results indicate that extracts of *Sargassum* spp. and *A*. *nodosum* are effective in mitigating high-temperature stress, making their use a promising alternative for inducing resistance in plants to the daily adversities of climate change.

## 1. Introduction

Among abiotic factors, temperature is one of the most important factors that significantly affects the physiological, morphological, and biochemical processes of plants [1,2]. Temperature stresses are normally classified into low- and high-temperature stress, which are below and above the optimal temperature range of plants [3].

Extreme temperatures today are a consequence of climate change, which is a reality that we have to face through scientific knowledge and technology, since at this time it represents a large-scale challenge for sustainable food production and for general human health [4]. With the process of climate change, it is inevitable that changes will continue to occur in traditional patterns of temperature, precipitation, and atmospheric humidity, which will constantly increase the number of negative scenarios for agriculture due to the greater intensity and fluctuation of climatic factors that are responsible for stress induction [5]. High temperatures are one of the main abiotic stresses that have detrimental impacts on the yield and growth of agricultural crops, where it has been reported that it may be responsible for up to 60% of yield losses [6].

High temperatures increase metabolic activity (high speed of vibration and translation of molecules) [3,7]. In addition to this, they cause a very high vapor pressure deficit in plants, which causes high transpiration, which can sometimes exceed the absorption of water by the roots, which results in stomatal closure, low photosynthetic activity, decreased growth, and burning of leaves, stems and fruits, causing low yields and poor quality of fruits [8,9]. At the molecular level, high temperatures compromise the functionality of biomolecules by reducing their catalytic capacity, or in the worst case, they can inactivate or denature them when the tolerance threshold is exceeded [10]. For terrestrial plants, the threshold where high temperatures begin to negatively affect biomolecules and the interaction processes between them is between 35 °C for those with C3 metabolism and 40 °C for those with C4 metabolism [3].

When plants are subjected to stress due to high temperatures, they activate the antioxidant system (enzymatic and non-enzymatic) to eliminate or neutralize the reactive oxygen species (ROS) produced, but sometimes the stress is greater and requires other measures to overcome it [11,12]. These situations have led to the search for alternatives to mitigate this type of stress, and among the most promising is the use of plant biostimulants. A biostimulant is any substance or microorganism applied to plants with the objective of improving nutritional efficiency, stress tolerance, and quality traits of crops, regardless of their nutrient content [13]. Among the most used biostimulants today are algae extracts, which represent a promising option due to the large amount of biomolecules they contain [14]. The most used extracts are those of brown algae (*Ascophyllum nodosum*, *Ecklonia maxima*, *Fucus* spp., *Lessonia nigrescens*, *Macrocystis pyrifera*, *Laminaria* spp., and *Sargassum* spp.) due to the high concentration of metabolites they contain such as carbohydrates, proteins, amino acids, phytohormones, carotenoids, vitamins, phenols, and inorganic compounds [15,16,17]. These metabolites have the ability to stimulate natural processes that improve nutrient absorption and assimilation, provide stress tolerance, and improve plant growth and yield [18].

The *Sargassum* spp. algae is currently causing different problems (ecological, tourist, and public health), due to atypical accumulations of this algae on Caribbean coasts [19,20]. Given this problem, different uses began to be given to it and one of them is in horticulture. The *Sargassum* spp. seaweed extracts have been used in horticultural plants to improve yield, quality, and give plants tolerance to different types of stress; however, mitigating stress due to high temperatures has not yet been studied. However, there are studies that show that its application can increase the synthesis of total phenolic compounds, flavonoids, carotenoids, and glutathione, compounds that can reduce ROS produced during stress [14,15,18]. This makes them a viable option to combat high-temperature stress in horticultural plants. In addition to this, there are some studies with other genera of brown algae that address this problem. In a study on tomato plants (*Solanum lycopersicum* L.), extracts of *A*. *nodosum* were shown to mitigate high-temperature stress by improving pollen viability, fruit yield, and the expression of genes encoding heat shock proteins (*HSP*) [11]. *A*. *nodosum* is among the most widely used algae in agriculture, as it has been shown to induce plants to produce their own hormones, which contributes to improving growth, mitigating stress, and improving the absorption and translocation of nutrients [21].

The mechanism of action of brown algae, which induces resistance to abiotic stress in plants, can have significant effects, such as reducing agrochemical use and lessening environmental impacts, in addition to addressing the problems of atypical *Sargassum* upwelling along coasts. In line with the above, the objective of the present investigation was to evaluate the effect of an extract of *Sargassum* spp. and *A*. *nodosum* on the induction of tolerance to high-temperature stress in tomato plants. The hypothesis was proposed that seaweed extracts would improve morphological, physiological, biochemical, and transcriptional aspects that would give tomato plants greater tolerance to high-temperature stress. The use of biostimulants derived from natural resources such as brown algae is promising, as they can help address the current challenges facing vegetable production.

## 2. Results

### 2.1. Plant Growth and Biomass

Table 1 shows the results of the height, stem diameter, and number of leaves of the tomato plants. For the height of the plants at 11 days after transplant (DAT), there were no differences. At 21 and 31 DAT, there were only differences in the non-stress treatments, where *Sargassum* spp. seaweed extract (SSE) plants exceeded control (AC) plants by 4.90% and 2.71%, respectively (*p* ≤ 0.05). For stem diameter and number of leaves, there were no differences for any of the two treatment groups (without and with stress) in any evaluation.

Figure 1 shows the results of the dry biomass of the plants. Regarding the aerial dry biomass in the treatments without stress, the SSE exceeded the AC by 9.56%, and in the treatments with stress, the SSE + 45 °C exceeded the AC + 45 °C by 6.66% (*p* ≤ 0.05). For the dry root biomass, there was no difference for any of the two treatments groups. For the total dry biomass, there were only differences for the non-stress treatments, where the SSE exceeded the AC by 8.58% (*p* ≤ 0.05). Figure 2 shows images of the tomato plants used in the experiment.

### 2.2. Stomatal Conductance

Regarding stomatal conductance, a significant decrease was observed in the stress treatments (*p* ≤ 0.05) (Figure 3), which is normal, since it is part of the plant’s defense mechanism. In non-stress treatments, a very marked difference was observed between the application of the extracts (*A*. *nodosum* commercial product (ANCP) and SSE) and the AC, where the latter presented the lowest values in the three evaluations. In this case, stress was applied only once, between 20 and 21 DAT, at which time the stressed treatments were equal, and a difference was found at 31 DAT, where ANCP + 45 °C and SSE + 45 °C were higher than AC + 45 °C by 58.34% and 63.57%, respectively (*p* ≤ 0.05).

### 2.3. Photosynthetic Pigments

Figure 4 shows the results of photosynthetic pigments, where significant differences between treatments are observed. For chlorophyll *a*, in the non-stress treatments, SSE increased the concentration by 33.76% (11 DAT), 26.66% (21 DAT), and 13.22% (31 DAT), compared to the AC (*p* ≤ 0.05). In the stress treatments there were only differences at 21 and 31 DAT, where the SSE + 45 °C group concentration exceeded that of the AC + 45 °C group by 22.05% and 12.38%, respectively (*p* ≤ 0.05). At 21 DAT the ANCP + 45 °C group concentration was also higher than that of the AC + 45 °C group (*p* ≤ 0.05).

Regarding chlorophyll *b*, there were only differences in the non-stress treatments, where the SSE group concentration was greater than the AC at 11 and 21 DAT by 12.40% and 11.96%, respectively (*p* ≤ 0.05).

For total chlorophyll, in the non-stress treatments, SSE increased the concentration by 23.71% (11 DAT), 19.84% (21 DAT), and 10.58% (31 DAT), compared to AC (*p* ≤ 0.05). In the stress treatments, SSE + 45 °C increased the concentration by 13.33% (21 DAT) and 7.45% (31 DAT), compared to AC + 45 °C (*p* ≤ 0.05).

### 2.4. Enzymatic Activity

The experiment also quantified the activity of ROS-scavenging enzymes such as superoxide dismutase (SOD), catalase (CAT), and ascorbate peroxidase (APX) (Figure 5). It is noticeable that the three enzymatic activities increase with stress and with the application of algae extracts. In this sense, the effects of applying SSE without stress was statistically higher than AC, and likewise the effect of SSE + 45 °C was higher than AC + 45 °C (*p* ≤ 0.05), for SOD, CAT, and APX in the three samplings (11, 21, and 31 DAT), which tells us about the biostimulant effect of SSE under standard growth conditions and low stress due to high temperatures. The ANCP had a similar effect to the SSE for CAT, where in conditions without stress and with stress, it was higher than AC and AC + 45 °C, respectively. ANCP application under stressful conditions caused a significant increase (*p* ≤ 0.05) in SOD at 11 and 21 DAT compared to AC + 45 °C. However, for APX, ANCP without stress was equal to AC and ANCP + 45 °C was equal to AC + 45 °C.

### 2.5. Total Proteins

In the treatments without stress, the effects of SSE in the three samplings was superior to the AC, and the effects of the ANCP only surpassed the AC at 21 and 31 DAT (*p* ≤ 0.05) (Figure 6). With respect to the treatments with stress, only at 31 DAT were there differences, with ANCP and SSE being superior to AC (*p* ≤ 0.05) (Figure 6).

### 2.6. Non-Enzymatic Antioxidants and Antioxidant Capacity

Table 2 shows the results for non-enzymatic antioxidants and antioxidant capacity in the leaves of tomato plants. For the content of total phenols, there were only differences at 21 and 31 DAT in the stressed treatments, where the SSE + 45 °C exceeded the AC + 45 °C by 18.17% and 19.33%, respectively (*p* ≤ 0.05).

For ascorbic acid, in the three samplings, ANCP and SSE were superior to AC in the treatments without stress, and likewise in the treatments with stress, ANCP + 45 °C and SSE + 45 °C were superior to AC + 45 °C (*p* ≤ 0.05).

For antioxidant capacity, in the treatments without stress, there were only differences at 11 DAT, with ANCP and SSE being superior to AC (*p* ≤ 0.05). For the treatments with stress, ANCP and SSE were superior to AC in all three samplings (*p* ≤ 0.05).

### 2.7. Expression of Defense Genes

Given the greater capacity of seaweed extracts to neutralize ROS by stimulating the enzymatic and non-enzymatic antioxidant systems, the transcript levels of nine stress-related genes (*NCED1*, *HSP70*, *PIP2*, *P5CS1*, *ERD15*, *Fe-SOD*, *CAT1*, *cAPX2* and *PAL5-3*) were examined (Figure 7).

It can be seen that in the first sampling at 11 DAT, where the plants were not yet subjected to stress, gene expression was somewhat dispersed among all treatments; however, at 21 and 31 DAT, when the plants had already been subjected to stress, gene expression spiked in the treatments with application of extracts and stress by high temperatures.

At 21 DAT, which was after stress and the third application of the extracts, the ANCP and SSE treatments under stressful conditions overexpressed most of the studied genes, with *HSP70*, *ERD15*, and *Fe-SOD* being the most notable. At 31 DAT, the use of the extracts under stressful conditions also increased the expression of the genes evaluated, with *NCED1*, *HSP70*, *P5CS1*, and *ERD15* being the most prominent. Overall, SSE under stress conditions induced increased expression of all genes at 21 and 31 DAT.

Furthermore, the application of the algal extracts under stress-free conditions, mainly at 21 and 31 DAT, repressed the expression of the genes *PIP2*, *P5CS1*, *Fe-SOD*, *CAT1*, *cAPX2*, and *PAL5-3*.

These transcriptional findings support the data observed through the physiological and biochemical analyses in the study and argue for a strong involvement of brown seaweed extracts in the activation of the defense system of tomato plants subjected to stress due to high temperatures.

## 3. Discussion

Currently, the use of biostimulants in agricultural crops is a recurrent practice due to the multiple benefits they provide to plants and the environment. Brown algae extracts, thanks to the large amounts of biomolecules they contain, are among the most promising biostimulants. The results of the present experiment show some improvements in growth, biomass, and stomatal conductance of tomato plants. Similar results were shown by Melo et al. [22], who in their research demonstrated that the use of extracts from a mixture of *Sargassum vulgare* and *Kappaphycus alvarezii* in bell peppers increased fruit production, plant growth and biomass, and stomatal conductance of leaves, all under standard growth conditions. Similarly, Repke et al. [23] indicate that the application of an extract of *A*. *nodosum* (1 L · ha^–1^) in soybeans under high-temperature stress improved growth parameters, yield, and stomatal conductance of the leaves, in addition to causing a significant reduction in leaf temperature.

These positive effects on plant growth, biomass, and stomatal conductance induced by brown algae extracts are mainly due to their contents of carbohydrates, proline, and glycine-betaine, which are osmolytes that help plants retain water under stress conditions, improving water regulation, photosynthesis and the development of foliar, floral, and root meristems [24]. In addition, brown algae extracts can stimulate carbon and nitrogen metabolism, which can improve plant growth and biomass [25].

High temperatures cause a very high vapor pressure deficit, which results in high transpiration that competes with the flow of water from other organs [26]. When transpiration exceeds water absorption, there is burning of leaves, stems, flowers, and fruits, as well as root damage, resulting in lower agronomic development and poor quality productions [27]. With stress from high temperatures, plants can eliminate certain parts of their organs or sometimes eliminate them completely [28]. This may be caused by burns or as a defense response to avoid having to consume energy to maintain these organs [3]. To do this, plants have a network of chain signals in their defense system that informs them about high-temperature stress, and in this way, levels of ethylene and abscisic acid (ABA), compounds responsible for organ senescence and abscission, are immediately elevated [11,29].

When high-temperature stress is prolonged and plant transpiration has already exceeded water absorption by the roots, plants close their stomata as a defense measure to avoid water loss as much as possible, which results in a decrease in stomatal conductance and a subsequent increase in leaf temperature [30]. However, brown seaweed extracts can improve this condition due to the considerable concentration of osmolytes they contain.

Figure 2 shows that the tomato plants used in the study under conditions of stress due to high temperatures began to lose part of their foliage; however, with the use of SSE and ANCP, this loss was significantly reduced.

Regarding pigments, it is observed that they increase with the application of brown algae extracts. Similar results are reported by Goyal et al. [31], who indicate that the use of *A*. *nodosum* extracts improved the photosynthetic rate and chlorophyll content in leaves of *Brassica juncea* (L.) Czern & Coss. under optimal growth conditions and high-temperature stress. Under standard growth conditions, Zermeño Gonzalez et al. [32] report an increase in chlorophylls in grapevine leaves with the use of *Sargassum* spp. extracts.

In this sense, it is common that the application of low doses of brown algae extracts increases the concentration of pigments in plants, and this is mainly due to glycine-betaine, which acts to protect the extrinsic protein structure of the photosynthetic complex, particularly in photosystem II [33]. In addition, glycine-betaine maintains photosynthetic activity by improving RUBISCO activity and chloroplast stability [33,34].

When high-temperature stress is prolonged, chlorophylls tend to degrade. This is due to the activation of enzymes such as chlorophyllases and peroxidases, which break down chlorophyll structures [35]. High temperatures also increase the synthesis of ROS, which affect chloroplast membranes and directly oxidize chlorophylls [36]. In addition, key genes in chlorophyll biosynthesis are repressed by high temperatures [37].

Among the effects that can be obtained with the application of brown algae extracts and high-temperature stress are the agronomic and photosynthetic parameters already mentioned; however, there is also an influence on the antioxidant system (enzymatic and non-enzymatic), where all these effects are a response to the expression of defense genes. The study showed that brown algae extracts increased the antioxidant system (enzymatic and non-enzymatic) and the expression of defense genes and in some cases there was repression. Repke et al. [23] report that foliar application of an *A*. *nodosum* extract to soybean plants under high-temperature stress increased the activity of SOD, CAT, APX, and peroxidase (POD) enzymes, as well as proline concentration, which assumes that the genes encoding these enzymes and proline were overexpressed. Similarly, Carmody et al. [11] evaluated foliar application of an *A*. *nodosum* extract on tomato crop under high-temperature stress and indicated that genes encoding for heat shock proteins (*HSP101.1*, *HSP70.9* and *HSP17.7C-CI*) were expressed. Mohammed et al. [38] evaluated the application of an extract of *Sargassum polycystum* on *Vicia faba* and *Helianthus annuus* seeds and indicated that the concentration of carotenoids, carbohydrates, total phenolic compounds, and flavonoids increased.

It is evident that high-temperature stress and the application of algal extracts influenced the antioxidant system and the expression of the genes evaluated. There are different routes through which the expression of defense genes can be achieved and the best known route is through membrane receptors with the ability to bind to the metabolites of the extracts (amino acids, phenols, carotenoids, carbohydrates, phytohormones, and inorganic compounds) and perceive thermal stress [39]. When this happens, the apoplastic Ca^2+^ and the one found in the vacuoles is transported to the cytoplasm and binds to proteins called calmodulins, forming a complex that activates protein kinases with the capacity to phosphorylate transcription factors, which travel to the nucleus to bind to specific DNA sequences, thus regulating gene expression [39,40]. This action allows the synthesis of proteins with antioxidant activity such as SOD, CAT, APX, glutathione peroxidase (GPX), etc., which are the first line of defense against stress by neutralizing ROS [41]. Proteins such as phenylalanine ammonia lyase (PAL), mitochondrial L-GalL dehydrogenase, and glutathione synthase, which synthesize phenolic compounds, ascorbic acid, and reduced glutathione, respectively, compounds that have the highest antioxidant activity in plant cells, can also be synthesized [42,43,44]. However, biomolecules contained in algal extracts can enter at the cellular level and directly initiate the signaling process for the subsequent expression of defense genes or act directly in the reduction in ROS [45].

The genes evaluated have primary functions in the plant defense system. The *NCED1* gene encodes for the enzyme involved in the synthesis of ABA, a phytohormone that plays a crucial role in the response to osmotic stress by regulating stomatal opening and closing [46]. The *HSP70* gene encodes for a heat shock protein whose function is to prevent misfolding and denaturation of other biomolecules under heat stress conditions [47]. The *PIP2* gene encodes for an aquaporin with the ability to regulate water transport in all plant organs [48]. The *P5CS1* gene codes for the enzyme that synthesizes the osmolyte proline, a compound of great importance in plant water regulation [49]. The *ERD15* gene encodes for an early response dehydration protein responsible for renaturation and protection of biomolecules exposed to thermal and osmotic stress [50].

The function of the enzymes encoding the *Fe-SOD*, *CAT1*, *cAPX2*, and *PAL5-3* genes were mentioned in the previous paragraph. In the experiment, the expression of the *Fe-SOD*, *CAT1*, *cAPX2*, and *PAL5-3* genes was quantified, showing a positive relationship between the enzymatic activity of SOD, CAT, and APX and the concentration of phenolic compounds related to the *PAL5-3* gene. This indicates that the antioxidant system in plants is largely mediated by the expression of the aforementioned genes.

The application of brown seaweed extracts to tomato plants not only promoted increased growth and biomass under standard growth conditions and high-temperature stress but was also accompanied by key physiological and biochemical improvements. Increased chlorophyll concentration is directly related to higher photosynthetic activity, which induces the synthesis of photoassimilates essential for plant growth and development [25]. Furthermore, increased activity of antioxidant enzymes (SOD, CAT, and APX), along with increased non-enzymatic antioxidants, provides the plants with a better ability to mitigate oxidative stress, helping to maintain cellular integrity and prolong tissue functionality [41]. Likewise, the overexpression of defense genes reflects a molecular response that underpins the observed physiological changes. Together, these integrated responses explain how stimulation with brown seaweed extracts strengthens both photosynthetic processes and defense mechanisms, resulting in increased growth and biomass accumulation in tomato plants subjected to high-temperature stress.

This study yielded promising results on how brown algae extracts can help plants mitigate stress caused by high temperatures. However, there were some differences between the extracts. First, in terms of growth variables, biomass, and stomatal conductance, the extracts performed similarly, but for photosynthetic pigments, enzyme activity, proteins, non-enzymatic antioxidants, and the expression of defense genes, the SSE demonstrated consistently superior effects to the ANCP. The greater effectiveness observed in the SSE can be attributed to the fact that it was applied immediately after extraction, which prevented the degradation of carbohydrates, amino acids, and secondary metabolites. This preservation of bioactive molecules made the physiological effect on tomato plants more efficient compared to commercial extracts, which have been stored for some time and may lose effectiveness. In this sense and considering the environmental impact caused by atypical accumulations of *Sargassum*, it can serve as a profitable and sustainable alternative to commercial products of *A*. *nodosum* and other types of synthetic products used in vegetable production.

Although the results demonstrate the potential of brown seaweed extracts to improve the tolerance of tomato plants subjected to high temperatures, it is important to note some limitations. First, the experiment was conducted over a short period of time (31 days), which may not fully reflect the extract’s long-term effects on plant growth, productivity, and physiological response throughout the crop cycle. Second, the study was carried out under controlled conditions, which, while allowing greater control over environmental variables, do not necessarily reproduce fluctuating field conditions; therefore, extrapolation of these results to real-life production scenarios should be carried out with caution. Finally, the chemical composition of brown seaweed extracts may vary depending on the species, geographical origin, and processing conditions, which could affect the reproducibility and consistency of the effects observed in different trials. Now, focusing on the defense gene expression data, four biological replicates were used per sampling, giving a total of 12 replicates divided into three samplings. However, this may be considered underpowered, especially for qPCR, which can be highly variable. Therefore, it is important to use a larger number of biological replicates.

## 4. Materials and Methods

### 4.1. Plant Matter and Experimental Conditions

The tomato seeds used were from the CID F1 hybrid of indeterminate growth (Harris Moran Seed Company, Modesto, CA, USA). The seeds were sown in polystyrene trays, using peat moss and perlite as substrate (1:1 *v*/*v*). The experiment was carried out in a greenhouse at the Horticulture Department of the Universidad Autónoma Agraria Antonio Narro (Saltillo, México). The average temperature was 28 °C at 60% relative humidity. The production of the seedlings in the tray took place over 35 days, until they developed four true leaves. Subsequently, the plants were transplanted into 1 L styrofoam containers, which contained peat moss and perlite in the same proportion used for sowing. Steiner [51] nutrient solution was used at 25% to nourish the plants.

An experiment was conducted to evaluate the effect of a *Sargassum* spp. seaweed extract (produced under different conditions described below) and a commercial extract based on *A*. *nodosum* in the induction of tolerance to high temperatures (45 °C) in tomato plants. A control with distilled water was used.

### 4.2. Treatments

The *Sargassum* spp. seaweed extract was produced in the Biorefinery pilot plant of the Universidad Autónoma de Coahuila (Saltillo, México), using a batch reactor, under the conditions of 160 °C, 30 min, and 50% ethanol. This extraction condition was selected through a preliminary test established by Sariñana-Aldaco et al. [14]. The ratio used for the extraction was 1 g of algae and 20 mL of 50% ethanol (1:20 *w*/*v*). The *Sargassum* spp. seaweed extract was applied foliarly at a dose of 1.5%; this dose was selected based on the reserach of Ramya et al. [52,53] and Kasim et al. [54]. The biochemical characterization of the *Sargassum* spp. seaweed extract is shown in the previous study reported by Sariñana-Aldaco et al. [55], where extracts produced under the same conditions were evaluated in tomato plants under salt stress. The quantified components of the extract were as follows: total proteins (3.47 mg g^–1^ DW), reduced glutathione (3.29 mg g^–1^ DW), amino acids (0.43 mg g^–1^ DW), total phenols (8.43 mg g^–1^ DW), flavonoids (2.83 mg g^–1^ DW), indole-3-acetic acid (0.57 mg kg^–1^ DW), trans-zeatin (175.99 µg g^–1^ DW), glucose (107.87 mg 100 g^–1^ DW), galactose (74.01 mg 100 g^–1^ DW), fucose (258.37 mg 100 g^–1^ DW), and mannitol (29.96 mg 100 g^–1^ DW).

The commercial extract of *A*. *nodosum* (BYOALG^®^, Mexico City, Mexico) was applied foliarly based on the recommendations for use (0.13%). This product is 100% *A*. *nodosum* according to the information presented on the label.

The treatments applied were the following: control with distilled water (AC), *A*. *nodosum* commercial product (ANCP), *Sargassum* spp. seaweed extract (SSE), AC and stress (AC + 45 °C), ANCP and stress (ANCP + 45 °C), SSE and stress (SSE + 45 °C), giving a total of six treatments with 16 replicates. The average constant temperature at which the plants in the stress-free treatments developed was 28 °C.

The treatments were applied foliarly and the plants were sprayed to the drip point. The applications occurred every 10 days from the transplant for a total of four applications during the test, which lasted 31 days. The third application of the treatments was 20 DAT and 12 h after the application; the plants were subjected to high-temperature stress (45 °C) for 12 h. The stress began at 20:00 h and ended at 08:00 h, so that the plants were not affected by the photoperiod. Only on this occasion was stress applied. In the tables and figures of results from 11 DAT and onwards, it is indicated which were the treatments that were subsequently subjected to stress. Laboratory incubators (Thermo 3310 Steri-Cult CO_2_ Double Incubator Unit3—AV, Marshall Scientific, Hampton, NH, USA) were used to subject the plants to high-temperature stress. By keeping the plants under stress in the incubator, the humidity remained close to 0%. Figure 8 shows schematically how the experiment was carried out.

### 4.3. Sampling and Evaluations

Three destructive samplings and three evaluations were carried out. At 10 DAT the second application of the treatments was carried out, and after 24 h, the first sampling and evaluation were carried out. At 20 DAT, the third application of the treatments was carried out, and after 12 h, the plants were subjected to stress for 12 h, where at the end of the stress, the second sampling and evaluation were carried out. At 30 DAT the fourth and final application of the treatments was carried out, and after 24 h, the third sampling and evaluation were carried out.

The destructive samplings consisted of four plants per treatment to quantify the activity and concentration of antioxidants (enzymatic and non-enzymatic); in addition to this, the expression of defense genes was also determined. Sampling was carried out by removing all the leaves from the plants and freezing them immediately with liquid nitrogen; they were subsequently stored in an ultra low temperature freezer at −80 °C. The evaluations consisted of four plants per treatment to evaluate height, number of leaves, stem diameter, and stomatal conductance of the leaves (SC-1 Leaf Porometer from METER Group). At the end of the experiment, the dry biomass of the plants was quantified.

### 4.4. Biomolecule Analysis

Analyses were performed on the leaves of tomato plants. To do this, the frozen tissue was freeze-dried and macerated with a hand mortar. Extractions and quantifications were performed on the dry powder of the tissue. The analysis of photosynthetic pigments was the only one performed with fresh tissue. The quantification of photosynthetic pigments was carried out using the methodology of Wellburn [56], which describes extraction with methanol and its subsequent reading in a UV-Vis spectrophotometer (Thermo Scientific Model G10S, Waltham, MA, USA) at wavelengths of 666 and 653 nm. The total protein content was determined following the methodology of Bradford [57], which describes extraction with phosphate buffer (0.1 M) and the use of Coomassie blue dye as a reaction agent for subsequent reading in a UV-Vis spectrophotometer at 595 nm and with the use of bovine serum albumin as a standard. The enzymatic activity of SOD (EC 1.15.1.1) was determined using a commercial Cayman^®^ 7060002 kit (Cayman Chemical Company, Ann Arbor, MI, USA) [58], and its results are expressed in U · mL^–1^, (U is defined as the amount of enzyme necessary to exhibit dismutation of 50% of the superoxide radical). The activity of the enzyme CAT (EC 1.11.1.6) was determined according to the methodology of Dhindsa et al. [59], and its results are reported as U · g^–1^ of total proteins, (U = equivalents in mM of H_2_O_2_ consumed in mL^−1^ · min^−1^). The activity of the enzyme APX (EC 1.11.1.11) was determined according to the methodology of Nakano and Asada [60], and its results are reported as U · g^–1^ of total proteins (U = μmol of oxidized ascorbate mL^–1^ · min^–1^). The three enzyme activities were quantified using a UV-Vis spectrophotometer.

Ascorbic acid was quantified by high-performance liquid chromatography (HPLC) (HPLC VARIAN 920LC), using the methodology of Nour et al. [61]. Total phenols were quantified with the methodology of Singleton et al. [62] through the Folin–Ciocalteu reaction, with the use of gallic acid as a standard and the use of a UV-Vis spectrophotometer at a wavelength of 750 nm. The antioxidant capacity was determined by the method described by Re et al. [63], which describes the use of ABTS as an oxidizing species and the use of ascorbic acid as a standard, using a UV-Vis spectrophotometer at a wavelength of 754 nm.

### 4.5. Real-Time Reverse Transcription PCR

For this analysis, fresh leaf tissue was used. TRI reagent (TRI Reagent^®^, Sigma-Aldrich, Burlington, MA, USA) was used to extract RNA from the leaves of tomato plants, which were subsequently purified with chloroform and precipitated with isopropanol, as described by Cui et al. [64]. The RNA was treated with DNase I (Sigma-Aldrich, Burlington, MA, USA) and quantified using a UV-Vis spectrophotometer with the A260/A280 nm ratio, and the quality was determined by denaturing electrophoresis. cDNA synthesis was performed using a commercial Bioline kit (SensiFAST cDNA Synthesis Kit, Bioline Reagents Ltd., London, UK). The primers were actin as an endogenous gene (*ACT*) and nine study genes: *NCED1* (9-cis-expoxycarotenoid dioxygenase 1), *HSP70* (heat shock protein 70), *PIP2* (plasma membrane intrinsic protein; aquaporin 2), *P5CS1* (delta1-pyrroline-5-carboxylate synthase 1), *ERD15* (protein of early response to dehydration 15), *Fe-SOD* (iron superoxide dismutase), *CAT1* (catalase 1), *cAPX2* (cytosolic ascorbate peroxidase 2), and *PAL5-3* (phenylalanine ammonia lyase 5-3). The primers were designed using Primer BLAST software (https://www.ncbi.nlm.nih.gov/tools/primer-blast/index.cgi?LINK_LOC=BlastHome, National Center for Biotechnology Information NCBI, Bethesda, USA) and Oligoanalyzer 3.1 (Integrated DNA Technologies IDT, Coralville, IA, USA), except for *ERD15*, *Fe-SOD*, and *cAPX2*, which were obtained from Ziaf et al. [65] and Mascia et al. [66], respectively. The sequences of the primers used are described in Table 3.

Real-time PCRs were performed on Applied Biosystems StepOne™ Equipment version 2.3 (Thermo Fisher Scientific, Waltham, MA, USA) using the ∆∆Ct method, measuring the fluorescence intensity of SYBR™ Select Master Mix (Applied Biosystems, Foster City, CA, USA). PCR was performed in a volume of 20 µL for all genes (10 µL Master Mix, 1 µL of cDNA, the concentration of primers, and nuclease-free water). For the actin gene, the concentration of the forward primer was 72 nM, and 60 nM was used for the reverse primer. For the *NCED1*, *ERD15*, *cAPX2*, and *Fe-SOD* genes, the concentrations of the primers were 300 nM equimolar. For the *HSP70* gene, the concentration of the forward primer was 80 nM, and that of the reverse primer was 100 nM. For the *PIP2* and *P5CS1* genes, the primer concentration was 100 nM equimolar. For the *CAT1* gene, the primer concentration was 200 nM equimolar. For the *PAL5-3* gene, the concentration of the forward primer was 150 nM and 100 nM for the reverse primer. Real-time PCR was run under the following conditions: 10 min at 95 °C and PCR (40 cycles): 15 s at 95 °C and 1 min at 60 °C.

### 4.6. Experimental Design and Data Analysis

The experimental design was completely randomized with a 3 × 2 factorial arrangement and 16 repetitions per treatment. The 16 replicates were divided as follows: four for the first destructive sampling, four for the second, and four for the third; the remaining four replicates were used to perform the evaluations of height, stem diameter, number of leaves, biomass, and stomatal conductance. A two-way analysis of variance was performed to observe the interaction of the two factors and their respective levels (extracts: distilled water, SSE and ANCP; stress: 28 and 45 °C) and a Fisher LSD test of means (*p* ≤ 0.05). Statistical analysis was performed using Infostat statistical software (v2020). Gene expression results are shown in heat maps, which were performed in GraphPad Prism 8.0 statistical software.

## 5. Conclusions

In this study, it was demonstrated that the application of extracts of *Sargassum* spp. and *A*. *nodosum* improved some growth and biomass parameters of tomato plants under optimal growth conditions and high-temperature stress. In addition, stomatal conductance, chlorophylls, antioxidants, and gene expression increased with the extracts under both growth conditions, with the *Sargassum* extracts being the most effective. The use of brown seaweed extracts in mitigating high-temperature stress is very limited, with only a few studies on *A*. *nodosum* and none on *Sargassum*, which represents a promising alternative for mitigating this type of stress in plants, as well as being a viable option for combating the problem of atypical *Sargassum* blooms on the Caribbean coast. Therefore, further studies are needed on the use of brown algae extracts in plants subjected to the aforementioned stress and to analyze other compounds such as ROS, phytohormones, osmolytes, minerals, and some other antioxidants and genes that indicate the state of plants.

## Figures and Tables

**Figure 1 plants-14-02996-f001:**
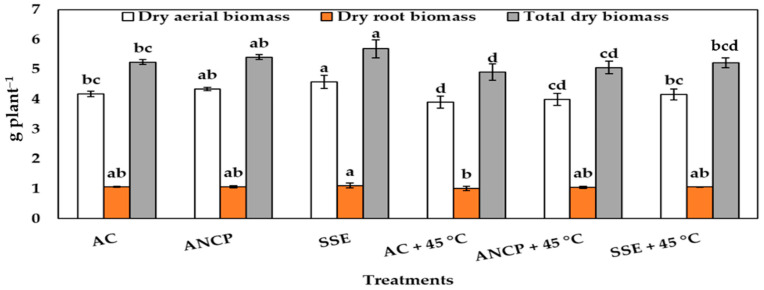
Dry biomass of the tomato plants at the end of the experiment. Different letters indicate significant differences between treatments (LSD, *p* ≤ 0.05). AC: control; ANCP: *A*. *nodosum* commercial product; SSE: *Sargassum* spp. seaweed extract; Values are presented as means ± SD, *n* = 4.

**Figure 2 plants-14-02996-f002:**
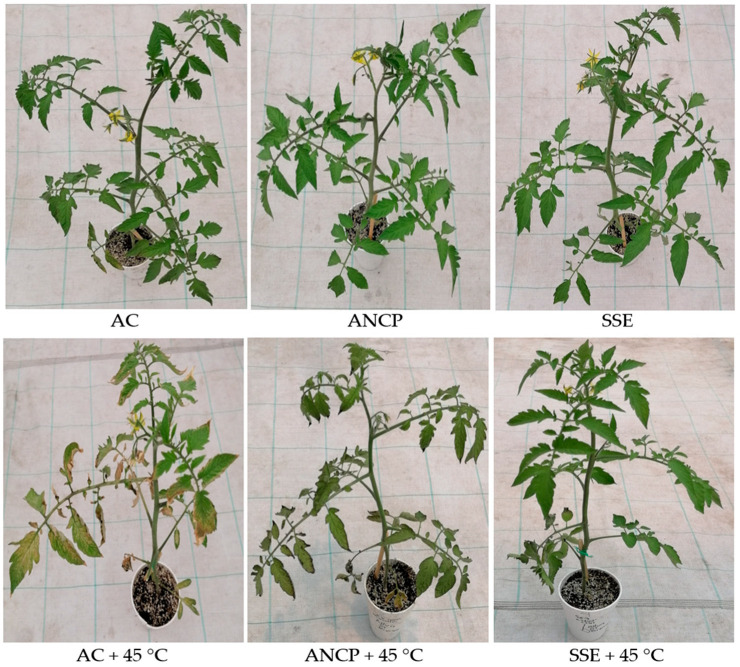
Tomato plants in the experimental high-temperature test at 31 DAT. AC: control; ANCP: *A*. *nodosum* commercial product; SSE: *Sargassum* spp. seaweed extract.

**Figure 3 plants-14-02996-f003:**
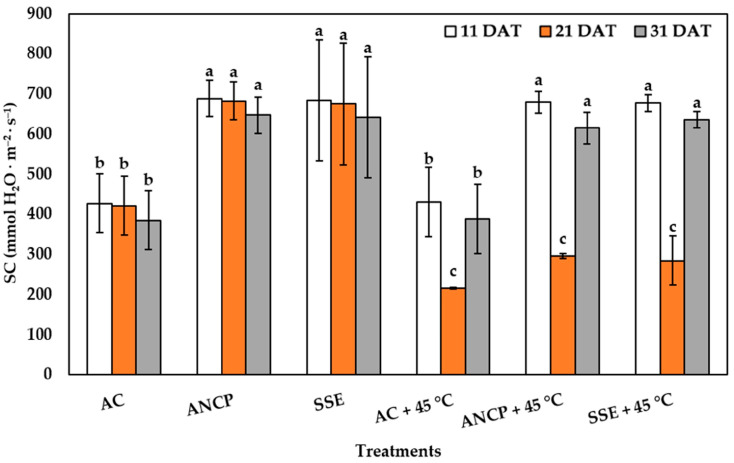
Stomatal conductance in the leaves of tomato plants. Different letters indicate significant differences between treatments (LSD, *p* ≤ 0.05). DAT: days after transplant; SC: stomatal conductance; AC: control; ANCP: *A*. *nodosum* commercial product; SSE: *Sargassum* spp. seaweed extract; Values are presented as means ± SD, *n* = 4.

**Figure 4 plants-14-02996-f004:**
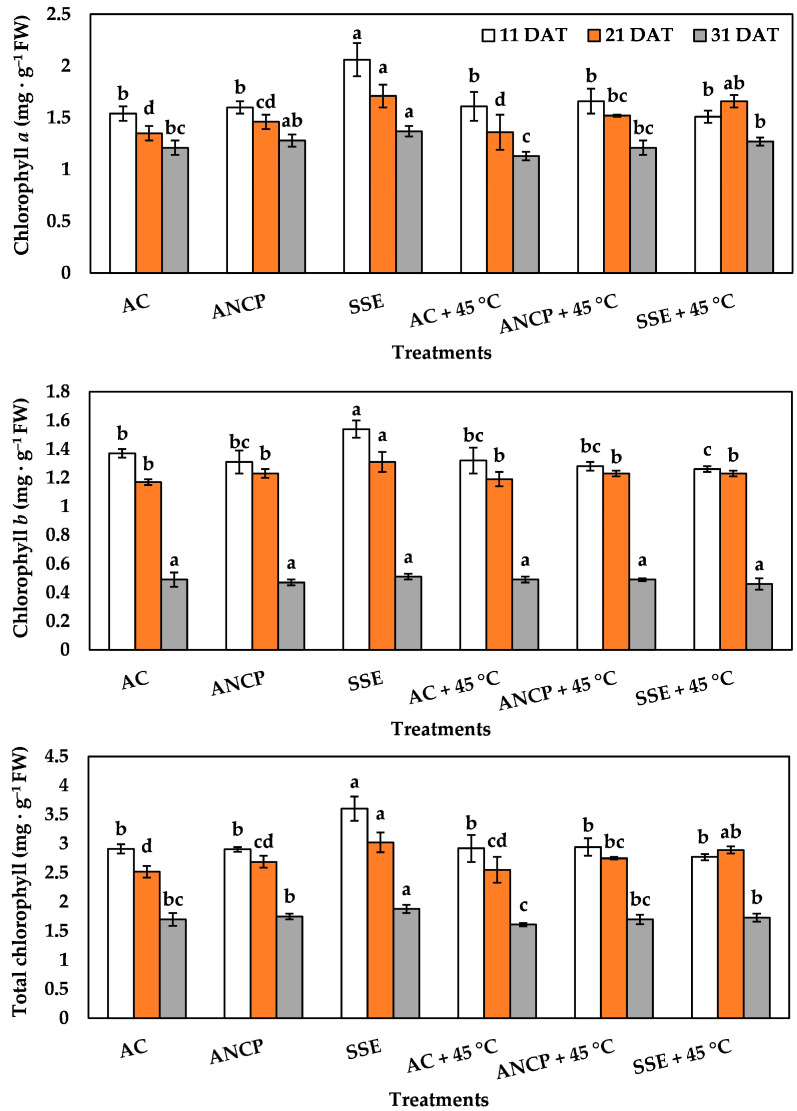
Photosynthetic pigments in the leaves of tomato plants. Different letters indicate significant differences between treatments (LSD, *p* ≤ 0.05). DAT: days after transplant; FW: fresh weight; AC: control; ANCP: *A*. *nodosum* commercial product; SSE: *Sargassum* spp. seaweed extract; Values are presented as means ± SD, *n* = 4.

**Figure 5 plants-14-02996-f005:**
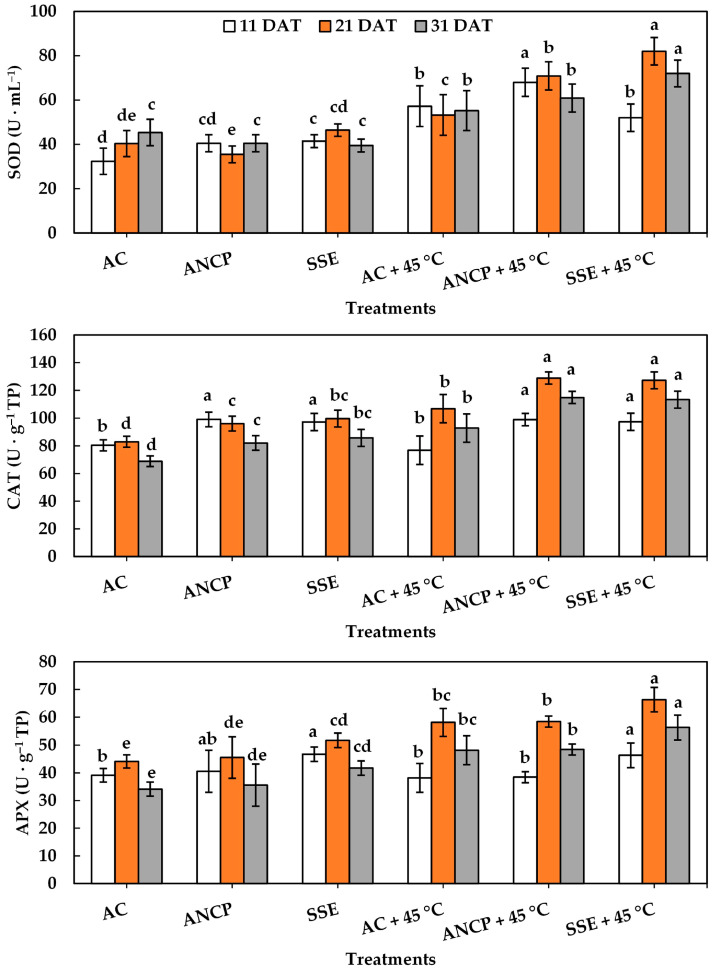
Enzymatic activity in the leaves of tomato plants. Different letters indicate significant differences between treatments (LSD, *p* ≤ 0.05). DAT: days after transplant; SOD: superoxide dismutase; CAT: catalase; APX: ascorbate peroxidase; AC: control; ANCP: *A*. *nodosum* commercial product; SSE: *Sargassum* spp. seaweed extract; Values are presented as means ± SD, *n* = 4.

**Figure 6 plants-14-02996-f006:**
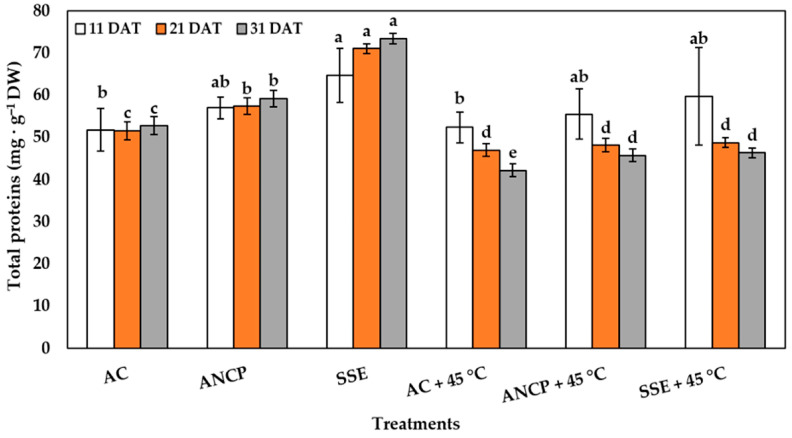
Total proteins in the leaves of tomato plants. Different letters indicate significant differences between treatments (LSD, *p* ≤ 0.05). DAT: days after transplant; DW: dry weight; AC: control; ANCP: *A*. *nodosum* commercial product; SSE: *Sargassum* spp. seaweed extract; Values are presented as means ± SD, *n* = 4.

**Figure 7 plants-14-02996-f007:**
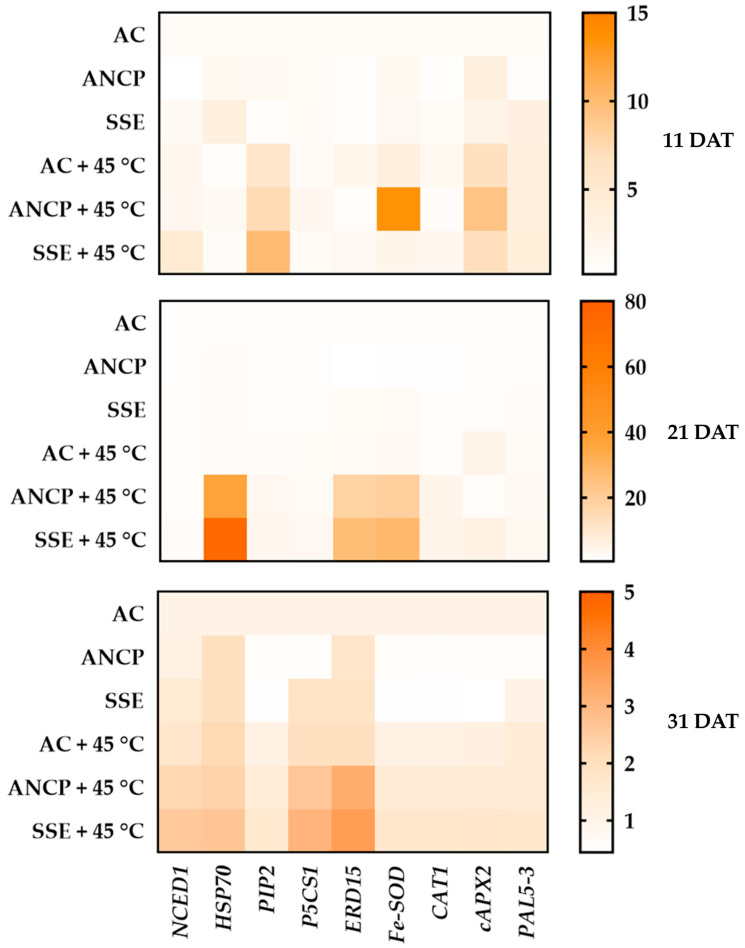
Heatmap of the relative expression of defense genes in leaves of tomato plants. DAT: days after transplant; AC: control; ANCP: *A*. *nodosum* commercial product; SSE: *Sargassum* spp. seaweed extract; *n* = 4. The AC represents a constant value of 1 at the expression level.

**Figure 8 plants-14-02996-f008:**
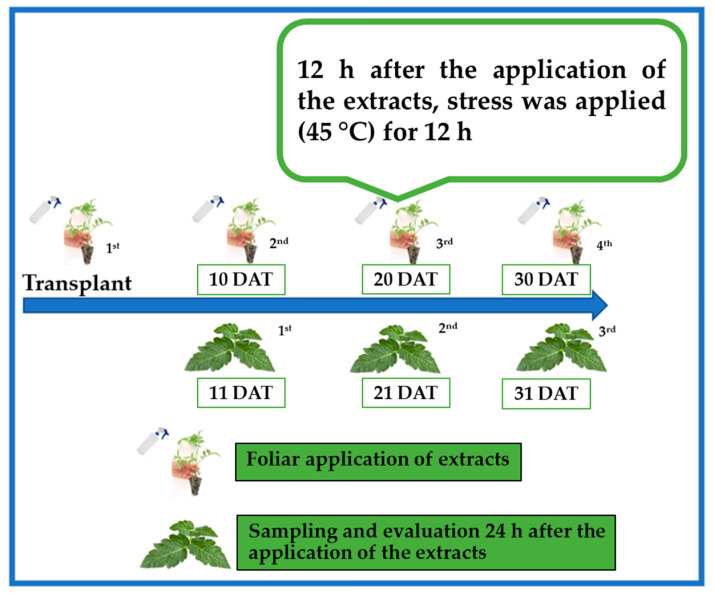
Schematic representation of the experiment. DAT: days after transplant.

**Table 1 plants-14-02996-t001:** Growth parameters of tomato plants.

Evaluation	Treatments	Plant Height (cm)	Stem Diameter (mm)	Leaf Number
	AC	12.08 ± 0.30 a	3.33 ± 0.17 a	6.75 ± 0.50 a
	ANCP	12.58 ± 0.47 a	3.40 ± 0.14 a	6.75 ± 0.50 a
11 DAT	SSE	12.65 ± 0.44 a	3.48 ± 0.09 a	6.75 ± 0.50 a
	AC + 45 °C	12.25 ± 0.47 a	3.30 ± 0.14 a	6.75 ± 0.50 a
	ANCP + 45 °C	12.53 ± 0.45 a	3.38 ± 0.09 a	6.75 ± 0.50 a
	SSE + 45 °C	12.55 ± 0.36 a	3.43 ± 0.18 a	7.00 ± 0.00 a
	AC	19.78 ± 1.04 bc	3.89 ± 0.18 abc	10.50 ± 0.57 a
	ANCP	20.40 ± 0.52 ab	3.96 ± 0.15 ab	10.75 ± 0.50 a
21 DAT	SSE	20.75 ± 0.36 a	4.04 ± 0.10 a	10.50 ± 0.57 a
	AC + 45 °C	19.10 ± 0.43 c	3.78 ± 0.09 c	10.25 ± 0.50 a
	ANCP + 45 °C	19.48 ± 0.28 c	3.80 ± 0.08 bc	10.25 ± 0.50 a
	SSE + 45 °C	19.43 ± 0.72 c	3.80 ± 0.08 bc	10.25 ± 0.50 a
	AC	35.78 ± 1.05 b	5.04 ± 0.17 a	14.50 ± 0.60 a
	ANCP	36.40 ± 0.50 ab	5.11 ± 0.15 a	14.75 ± 0.50 a
31 DAT	SSE	36.75 ± 0.40 a	5.19 ± 0.09 a	14.50 ± 0.60 a
	AC + 45 °C	34.10 ± 0.40 c	4.78 ± 0.10 b	14.25 ± 0.50 a
	ANCP + 45 °C	34.48 ± 0.30 c	4.83 ± 0.05 b	14.25 ± 0.50 a
	SSE + 45 °C	34.43 ± 0.70 c	4.80 ± 0.09 b	14.25 ± 0.50 a

Note: Different letters within each column indicate significant differences between treatments (LSD, *p* ≤ 0.05). DAT: days after transplant; AC: control; ANCP: *A. nodosum* commercial product; SSE: *Sargassum* spp. seaweed extract; *n* = 4; ± standard deviation (SD). The average temperature at which the experiment was conducted was 28 °C and was constant for the unstressed treatments.

**Table 2 plants-14-02996-t002:** Non-enzymatic antioxidants and antioxidant capacity.

Sampling	Treatments	Total Phenols (mg GAE g^–1^ DW)	Ascorbic Acid (mg 100 g^–1^ DW)	Antioxidant Capacity (mg AAE g^–1^ DW)
	AC	15.72 ± 1.42 a	49.66 ± 1.45 c	18.29 ± 1.50 c
	ANCP	16.11 ± 0.64 a	62.06 ± 4.20 b	20.36 ± 0.80 b
11 DAT	SSE	16.34 ± 1.15 a	70.13 ± 2.60 a	22.88 ± 0.74 a
	AC + 45 °C	15.77 ± 0.85 a	50.26 ± 2.70 c	18.13 ± 0.89 c
	ANCP + 45 °C	15.69 ± 0.75 a	61.54 ± 2.00 b	20.68 ± 0.70 b
	SSE + 45 °C	15.77 ± 0.73 a	70.04 ± 3.00 a	22.42 ± 1.28 a
	AC	12.59 ± 1.40 c	55.99 ± 4.28 e	17.75 ± 1.48 d
	ANCP	12.88 ± 0.65 c	70.60 ± 3.14 d	18.62 ± 0.67 d
21 DAT	SSE	13.22 ± 1.20 c	68.47 ± 5.49 d	19.19 ± 0.52 cd
	AC + 45 °C	17.33 ± 0.90 b	94.46 ± 5.50 c	20.30 ± 0.90 c
	ANCP + 45 °C	17.78 ± 0.80 b	161.17 ± 7.97 a	22.86 ± 0.71 b
	SSE + 45 °C	20.48 ± 0.70 a	132.86 ± 4.28 b	25.68 ± 1.30 a
	AC	11.55 ± 1.40 c	99.13 ± 4.30 e	16.66 ± 1.40 d
	ANCP	1184 ± 0.60 c	113.74 ± 3.15 d	17.53 ± 0.70 d
31 DAT	SSE	12.18 ± 1.10 c	111.61 ± 5.50 d	18.10 ± 0.50 cd
	AC + 45 °C	16.29 ± 0.80 b	137.61 ± 5.50 c	19.21 ± 0.89 c
	ANCP + 45 °C	1673 ± 0.70 b	204.31 ± 8.00 a	21.77 ± 0.71 b
	SSE + 45 °C	19.44 ± 0.72 a	176.00 ± 4.30 b	24.60 ± 1.28 a

Note: Different letters within each column indicate significant differences between treatments (LSD, *p* ≤ 0.05). DAT: days after transplant; AC: control; ANCP: *A*. *nodosum* commercial product; SSE: *Sargassum* spp. seaweed extract; GAE: gallic acid equivalents; AAE: ascorbic acid equivalents; DW: dry weight; *n* = 4; ± SD.

**Table 3 plants-14-02996-t003:** Primer sequences of the analyzed genes.

Gene	Forward Primer 5′-3′	Reverse Primer 5′-3′
*ACT*	CCCAGGCACACAGGTGTTA	CAGGAGCAACTCGAAGCTC
*NCED1*	CTTATTTGGCTATCGCTGAACC	CCTCCAACTTCAAACTCATTGC
*HSP70*	TGCTGGAGGTGTTATGACCA	GACTCCTCTTGGTGCTGGAG
*PIP2*	CTGCACCGTTGCTCGATTTT	GCGACAGTGACGTAGAGGAA
*P5CS1*	CTGTTGTGGCTCGAGCTGAT	GACGACCAACACCTACAGCA
*ERD15*	AGGCATCAAGTCATCACTCTCTGGT	GAGGTAAATGTGAGTAAGAACCAACG
*Fe-SOD*	CTGGGAATCTATGAAGCCCAACGGA	CAAATTGTGTTGCTGCAGCTGCCTT
*CAT1*	TCGCGATGGTGCTATGAACA	CTCCCCTGCCTGTTTGAAGT
*cAPX2*	GTGACCACTTGAGGGACGTGTTTGT	ACCAGAACGCTCCTTGTGGCATCTT
*PAL5-3*	GGAGGAGAATTTGAAGAATGCTGTG	TCCCTTTCCACCACTTGTAGC

## Data Availability

The data that support the findings of this study are available from the corresponding author, [S.G.-M.], upon reasonable request.

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
