# Peer review of "Brown Algae Extracts Increase the Tolerance of Tomato Plants to High Temperatures by Improving Morphological, Physiological, Metabolomic, and Transcriptional Parameters"

_plants, 2025, doi:10.3390/plants14192996_

Round 1
Reviewer 1 Report
Comments and Suggestions for Authors
Reviewer’s comment:
The manuscript titled “Brown algae extracts increase the tolerance of tomato plants to high temperatures by improving morphological, physiological, metabolomic, and transcriptional parameters” presents a well-designed study addressing the effects of Sargassum spp. and Ascophyllum nodosum extracts on tomato plants exposed to high-temperature stress. The study follows a multi-disciplinary approach, exploring the morphology, physiology, biochemistry, and gene expression. These pave the way to understanding the strong indicators of brown algae extracts as a potential biostimulant. The novelty in this study demonstrated that Sargassum spp., an underutilized resource causing environmental issues, can enhance heat tolerance in tomato, comparable to or exceeding the widely used A. nodosum.
The study is scientifically sound, and the results are promising; however, the manuscript would benefit from moderate revision to improve clarity and readability, reduce redundancy, sharpen the focus on novelty, and strengthen practical implications. With these revisions, the paper will make a valuable contribution to the field of plant stress physiology and sustainable biostimulant applications.
Abstract: Overall, the abstract was well-written with a concise overview of the study objectives, methods, and results. However, the final statement can be modified to the relevance of the results to real-time applications in horticulture.
Lines 31 – 39: this part of the abstract describing the results could be made more readable by improving the sentence structure or using simple sentences instead of complex sentences. Since the results for both stress and stress-free conditions share most similarities, the use of “however” in line 37 should be replaced and only used to describe the contrast in gene expression and repression.
Introduction
Comprehensively written, but with a bit more focus on the effect of climate change.
Lines 88 – 96: This section of the introduction that describes the knowledge gap should be rewritten for better understanding. While I was able to understand that there have been studies for A. nodosum and none for Sargassum sp. for heat stress in Tomato, the paragraph was difficult to read. “It was proven that A. nodosum extracts in tomato mitigated high temperature stress…” could be written as “In a tomato study, it was proven that A. nodosum extracts mitigated high temp stress…
Line 101: “This action can have significant effects…” which action are the authors referring to? The paragraph ending in line 100 does not have a good flow into the concluding paragraph in the introduction section.
Lines 101 – 103: This sentence should be rewritten for better clarity, and preferably use simple sentences to explain the benefits of the application of Sargassum in horticulture.
Lines 103 – 105: The objective of the present investigation “is”. Also, could a hypothesis or research question be included in this paragraph?
Materials and Methods
Please let the “materials and methods” section precede the “results” section.
This section described the experimental design, extract procedure, treatments, replicates and assays carried out in this study. This makes it extensive however, the details were very difficult to read and understand. The sentences were unnecessarily complex, some part verbose (extraction details).
Lines 499: “the place where the experiment was carried out was in a…” should be rewritten as “the experiment was carried out in a greenhouse…” this is more concise.
Line 532: 20 dat should be rewritten as 20 DAT, as this is an acronym. Also justify how the stress condition (45℃ for 12h – one time exposure) mimics the field heat stress condition.
Line 535 – 539: the sentence “however, in the tables and figures of result from the 11 dat…” should be reviewed for deletion, while the section should be rewritten concisely and for clarity
Lines 540 – 541: The schematic representation of the experiment timeline and treatment application was a great addition. However, the figure needs to be improved, detailed and clearly show the entire experiment in a good flow.
Lines 543 – 549: DAT not dat
Line 562: Pigment not pygment
Lines 567 – 574: the “U” in the units are not consistent. Not defined in total protein in line 567 while in line 571, it is consumption in mM of H2O2 but oxidation in µmol of ascorbic acid per minute in lines 573-574. What is U?
Lines 619 – 627: The data analysis used in the study was unclear. The authors mentioned a 3 x 2 factorial design but only mentioned that an analysis of variance and Fisher’s LSD for mean comparison was used. What type of ANOVA was used to check the interaction of effects between the factors?
Results
The section was well structured into sub-sections and the use of figures and tables made gave more clarity on the results. However, the results need to be summarised with only mentions of the numerical trends/% changes, since the numbers are clear on the table.
Line 110: Please note that the acronym dat is corrected across the document as DAT
Line 112: 4.90%
Line 115: Table 1 (21 DAT) the stem diameter for AC treatment has the “a-c”. please correct this.
At 21 DAT and 31 DAT, the mean values of stem diameter had different letters and according to the footnote means significant differences, yet it was reported in lines 113 – 114 that there were no differences in stem diameter in any evaluation. This is not clear, please explain.
Lines 174 – 185: Too many % increase reported without a good highlight on the trends
Discussion
The section had a broad literature backing to compare algae biostimulants and highlights the novelty of Sargassum extracts for heat stress. However, the comparison between Sargassum and A. nodosum should be more extensive. What does this study show? which is consistently superior? Were they complementary in their effects?
Lines 428: However, brown algae extract…” this sentence should be rewritten. The use of the words “thanks to” in an academic reporting style is weak.
Lines 492 – 494: This sentence should be properly linked with the previous paragraph. It is hanging and open-ended as the final sentence in the discussion section. In addition to the well-discussed gene expression, please tie these outcomes directly to the physiological changes that were experienced. How does CAT1 expression align with the enzymatic activities?
Conclusion
Lines 635 – 638: Sounds too generic and complex. It should be simplified to summarize the objective in a sentence and then explain the potential use of Sargassum as an alternative to the commercially sold A. nodosum products for heat stress in tomato plants while managing the environmental issue it causes.
Author Response
Dear reviewer, thank you for your comments aimed at improving the quality of the manuscript.
Responses to each comment are included in the attached file.

Reviewer 2 Report
Comments and Suggestions for Authors
Dear Authors,
I think that your manuscript presents a valuable investigation into brown algae-derived biostimulants and their potential to enhance plant resilience, contributing to sustainable crop management and stress mitigation strategies. I have provided my suggestions in the attached document, organized by sections, and I hope they will help you improve the manuscript.

Author Response

(The authors gave the same response as above.)

Round 2
Reviewer 1 Report
Comments and Suggestions for Authors
The authors have revised the paper substantially and only found minor grammatic errors which i believe can be address during proofreading.
Comments on the Quality of English LanguageThe authors have revised the paper substantially and only found minor grammatic errors which i believe can be address during proofreading.
Reviewer 2 Report
Comments and Suggestions for Authors
Dear Authors,
I have carefully reviewed your revised manuscript and am pleased to note that you have responded thoroughly to my earlier suggestions. The changes you have made have improved the quality and clarity of the work, and I believe it is ready for publication in Plants.